# The Improved Biocontrol Agent, F1-35, Protects Watermelon against Fusarium Wilt by Triggering Jasmonic Acid and Ethylene Pathways

**DOI:** 10.3390/microorganisms10091710

**Published:** 2022-08-25

**Authors:** Xiao-Min Dong, Qing-Gui Lian, Jing Chen, Rui-Min Jia, Zhao-Feng Zong, Qing Ma, Yang Wang

**Affiliations:** College of Plant Protection, Northwest A&F University, Yangling, Xianyang 712100, China

**Keywords:** watermelon Fusarium wilt, biological control, JA and ET pathway, proteomes, qRT-PCR

## Abstract

Watermelon Fusarium wilt, caused by *Fusarium oxysporum* f. sp. *niveum* (FON), is one of the most important diseases, and has become a major limiting factor to watermelon production worldwide. Previous research has found that the improved biocontrol agent, F1-35, had a high control efficiency to watermelon Fusarium wilt. In this study, the control efficiency of F1-35 to watermelon Fusarium wilt was firstly tested, and the control efficiency was 61.7%. Then, we investigated the mode of action of F1-35 in controlling watermelon Fusarium wilt. Using a pairing assay, we found that F1-35 did not inhibit the normal growth of FON. To know more about the interaction between F1-35 and watermelon root, the protein expressions of roots after 12, 24, and 48 h post-inoculation were examined. A total of 1109 differentially expressed proteins were obtained. KEGG analysis found that the most differentially expressed proteins occurred in alpha-linolenic acid metabolism, cysteine and methionine metabolism, plant–pathogen interaction, and the MAPK signaling pathway to the plant. A further analysis of differentially expressed proteins showed that F1-35 triggered the jasmonic acid and ethylene pathways in watermelon. To validate our results, the qRT-PCR was used to analyze the gene expression levels of *PAL*, *LOX1*, and *CTR1*. The gene expression results showed that those genes, which were positive correlated with the JA pathway, were up-expressed, including *PAL* and *LOX1*, and the negative associated gene, *CTR1*, was down-expressed. In conclusion, the improved biocontrol agent, F1-35, improves the resistance of watermelons to FON by triggering the JA and ET pathways.

## 1. Introduction

Fusarium wilt of watermelons (*Citrullus lanatus*), caused by *Fusarium oxysporum* f. sp. *niveum* (FON), is one of the most devastating diseases of watermelons around the world [1,2]. The pathogen which causes Fusarium wilt of watermelons exists in many different habitats, including saline soil, dry-tropical rainforests, and hospital sewage [3,4,5,6], and produces microspores, macrospores, and chlamydospores [7].

Fungicides, especially some fumigants, were once the most efficient way to control Fusarium wilt, but those fungicides were banned because of their defects, including being unfriendly to the environment and hardly degradable [8,9]. According to the Montreal Protocols, some alternative soil fumigants are used to control Fusarium wilt, but they are not as effective as MeBr at controlling FON or weeds, they are expensive, and they also negatively impact the environment [10]. Grafting is another method to control watermelon Fusariun wilt, which was first used commercially on watermelons in Japan and Korea [11,12]. Watermelons are primarily grafted onto *Cucurbita* rootstocks, which are non-hosts to FON [13,14,15]. However, as Mahamed [16] reported, grafting watermelons influences the growth, productivity, and quality of the fruit. Consequently, it is necessary to find an environmentally friendly, high-efficiency method to control Fusarium wilt.

Some researchers reported that *F. oxysporum*, as a hemi-biotrophic pathogen, could be inhibited by jasmonic acid and ethylene: the interaction between *F. oxysporum* and *Arabidopsis thaliana* reveals that salicylic acid (SA), which connects to systemic acquired resistance (SAR) [17,18], and JA/ET, which are induced systemic resistance (ISR) response hormones to beneficial microorganisms [19], also take part in the resistance process to *Fusarium* [20,21,22]. In addition, Kasote et al. [23] also found that ME-JA and JA-Ile may play an important role in the watermelon defense response against FON pathogens. Interestingly, some *F. oxysporum* species can enter into a plant as a beneficial microorganism and stimulate the JA pathway in the early stages [18,24]. As such, JA and ET play a role of resistance in the interaction between plant and *F. oxysporum*.

Many studies over the past decades have shown that certain microorganisms interact intimately with plants, suppressing plant pathogens and promoting plant growth [25,26,27,28]. Those microorganisms able to suppress plant pathogens are called biological control agents (BCA). The antagonism of BCAs includes: (1) diffusible antibiotics, volatile organic compounds, toxins, and biosurfactants inhibiting microbial growth; (2) competition space and nutrients; (3) degradation toxins and other pathogenicity factors; and (4) hyperparasitic, secreting cell-wall-degrading enzymes, such as chitinases and β-1,3-glucanase [29,30,31]. At present, some BCAs have been reported to have inhibitory effects on Fusarium wilt, including *Bacillus velezensis*, *Fluorescent pseudomonads*, *Trichoderma saturnisporum*, and non-pathogenic *F. oxysporum* [32,33,34,35].

Furthermore, some BCAs induce the expression of plant hormones, such as SA, ET, and JA [36,37,38,39]. Those plant hormones suppress infection by plant pathogens by plant hormone signal transduction. SA, JA, and ethylene (ET) are the major phytohormones in SAR and ISR, respectively. SAR and ISR are the important strategies by which plants respond to a wide range of biotic and abiotic stresses [40,41].

One of the BCAs, non-pathogenic *Fusarium oxysporum* Fo47, was isolated from soil in France [42]. Many researchers have reported that it has the ability to control many plant diseases worldwide, including Fusarium wilt, *Verticillium*, and *Phytophthora* in pepper and tomatoes [42,43,44,45,46,47]. In former research, protoplast fusion was carried out between Fo47 and actinomycetes 153 by inactivation and mutation in order to improve the characteristics of the Fo47 strain [48]. F1-35, one of the fusion products, improved the properties of inoculation, fungicide resistance, and growth promotion. We also found that the rates of promoting the growth of F1-35 depended on the dose of gibberellin. The recombinant strain, F1-35, produced a higher content of gibberellin (223.8 mg/L) than Fo47 (134.5 mg/L) after being separately grown in 200 mL Czapek broth for 3 days in a 250 mL flask at 25 °C, agitated at 150 rmp. F1-35 also had better growth-promoting than that of Fo47 in watermelons and cucumber seedlings. With regard to the control efficiency of FON, F1-35 reached 59.04%, which was higher than that of Fo47 (34.94%) [48].

The aim of this study was to determine the mechanism of F1-35 control of watermelon Fusarium wilt by using the pairing assay and isobaric tags for relative and absolute quantitation (iTRAQ) coupled with the LC-MS/MS method and qRT-PCR to study the interaction between F1-35 and watermelon plants.

## 2. Materials and Methods

### 2.1. Pathogenic Fungus, Biocontrol Agent F1-35, and Plant Materials

The watermelon Fusarium wilt pathogen, *Fusarium oxysporum* f. sp. *niveum* (FON), and the improved biocontrol strain, F1-35, were provided by the Vegetable Disease and Biocontrol Laboratory, College of Plant Protection, Northwest A&F University, Yangling, Shaanxi Province, China. FON and F1-35 were cultured on potato dextrose agar (PDA) plates at 28 °C for 7 d. A cultivated variety of watermelon, NW-5, which is sensitive to watermelon Fusarium wilt, was used in this study; this was provided by Associate Professor Jianxiang Ma, College of Horticulture, Northwest A&F University, Yangling, Shaanxi Province, China. The seeds were treated according to the procedures of Kong et al. [49] as follows: seeds were first sterilized with 1.5% sodium hypochlorite, soaked in distilled water for 4 h, and maintained at 30 °C for germination. The germinated seedlings were planted in a sterilized peat-perlite substrate (2:1 *v*/*v*) and cultured in the greenhouse under a 16-h diurnal light cycle at 28 °C with 80 to 85% relative humidity inside a controlled environment chamber. Seedlings with five true leaves were used for the experiment.

### 2.2. Inoculum Preparation and Plant Inoculation

Agar plugs (5-mm diameter) were aseptically cut out from the edge of the fungal colonies of FON and F1-35 after 7 d of growth in PDA plates. FON and F1-35 were then separately grown in 200 mL potato dextrose broth (PDB) in a 250 mL flask at 28 °C for 7 d, with agitation at 160 rmp. Spore suspensions of FON and F1-35 in PDB were filtered and prepared at concentrations of 10^6^ cfu/mL and 10^8^ cfu/mL, respectively, determined using a hemocytometer [50].

All seedling roots were inoculated by dipping in a spore suspension for 10 min. There were four treatment groups: control (CK) (treated with PDB), FON (treated with FON), F1-35 (treated with F1-35), and F1-35/FON (treated with FON and F1-35 simultaneously). Each treatment was conducted in triplicate, with 24 seedlings in each replicate. Samples of CK, FON, and F1-35 were separately collected at 0, 12, 24, 48, 72, and 96 h post-inoculation (hpi), with six seedlings for each time point, immediately frozen in liquid nitrogen and stored at −80 °C for further study.

### 2.3. Control Efficiency Test

Following the treatments described above, and with three replicates for each treatment, the disease index and control efficiency were counted using the methodology described by Pu et al. [50].

### 2.4. Pairing Assay

To determine the F1-35 inhibition of FON, dual cultures were used according to the method of Veloso and Díaz [43], including only FON and F1-35 + FON, with three replicates, each replicate comprising three Petri dishes. The mycelia of FON, which were located beside F1-35, were examined under a microscope.

### 2.5. Proteomic Analysis

#### 2.5.1. Proteome Extraction and iTRAQ Labeling

Protein from the CK, and 12 hpi, 24 hpi, and 48 hpi root of F1-35 samples, comprising three biological replicates, was extracted using TCA/acetone precipitation and SDT lysis [51]. Proteins (20 µg) from each sample were then mixed with 5× loading buffer, and boiled for 5 min. The proteins were separated on 12.5% SDS-PAGE gel (constant current 14 mA for 90 min). Protein bands were visualized by Coomassie Blue R-250 staining. After filter-aided sample preparation (FASP digestion) using the Wisniewski method [52], 100 μg of peptide mixture from each sample was labeled using the iTRAQ reagent according to the manufacturer’s instructions (Applied Biosystems).

#### 2.5.2. Peptide Fractionation with Strong Cation Exchange (SCX) Chromatography

The labeled peptides in each group were homogenized and mixed with an AK-TA purifier system (GE Healthcare). As buffer A, we used 10 mM KH_2_PO_4_, 25% ACN, pH 3.0, whereas buffer B contained 10 mM KH_2_PO_4_, 500 mM KCl, 25% ACN, pH 3.0. The column was equilibrated with buffer A, and then, peptides were eluted at a flow rate of 1 Ml/min with a gradient of 0–8% buffer B (500 mM KCl, 10 mM KH_2_PO_4_ in 25% of ACN, pH 3.0) for 22 min, 8–52% buffer B for 22–47 min, 52–100% buffer B for 47–50 min, and 100% buffer B for 50–58 min; then, buffer B was reset to 0% after 58 min. The absorbance was set at 214 nm for monitoring elution, and fractions were collected every 1 min. After collecting fractions, the fractions were collected on C18 Cartridges (Empore™ SPE Cartridges C18, standard density, bed I.D. 7 mm, 3-mL volume; Sigma, St. Louis, MO, USA) and vacuum-centrifuged for concentration.

#### 2.5.3. LC-MS/MS Analysis

The specific steps are as follows: (1) buffer A (0.1% formic acid) and buffer B (84% acetonitrile and 0.1% formic acid) were prepared. (2) The peptide mixture was loaded onto a reverse phase trap column (Thermo Scientific Acclaim PepMap100, 100 μm × 2 cm, nanoViper C18, Boston, MA, USA) and connected to a C18-reverse phase analytical column (Thermo Scientific Easy Column, 10 cm long, 75 μm inner diameter, 3μm resin). Buffer A and buffer B were used for elution at a flow rate of 300 nL/min, which was controlled by IntelliFlow technology. (3) The samples were analyzed by a Q Exactive mass spectrometer (Thermo Scientific), and the detection mode was set to positive ion for 60/120/240 min. The parent scan was in the range of 300 to 1800 *m*/*z* with a resolution of 70,000 at 200 *m*/*z*. The automatic gain control (AGC) target was set to 3e6 at a maximum injection time of 10 ms. The duration of dynamic exclusion was 40.0 s. The parent ions with the top 10 ionic strength in the full scan were selected and broken by high-energy collisional dissociation (HCD) fragmentation at a resolution of 17,500 at 200 *m*/*z*. The collision energy was normalized to 30 eV, and the underfill ratio was 0.1%.

#### 2.5.4. MS Data Analysis

MS/MS spectra were searched using the MASCOT engine (Matrix Science, London, UK; version 2.2) embedded into Proteome Discoverer 1.4 [53]. Protein data, uniprot_Benincaseae_27316_20170228.fasta, were downloaded on 28 February 2017 at http://www.uniprot.org. The following parameters were set: enzyme as trypsin, max missed cleavages at 2, fixed modification as carbamidomethyl (C), iTRAQ 4/8 plex (N-term) and iTRAQ 4/8 plex (K), variable modifications as oxidation (M) and iTRAQ 4/8 plex (Y), peptide mass tolerance at ±20 ppm level, fragment mass tolerance as 0.1 Da, Benincaseae as database, decoy as database pattern, peptide FDR at ≤0.01 level, and the protein ratios calculated as the median of only the unique peptides of the protein. All peptide ratios were normalized by the median protein ratio; the median protein ratio should, therefore, be 1 after normalization.

#### 2.5.5. Analysis of Differentially Expressed Proteins

Compared to CK, proteins with a differential expression greater than ±1.2-fold and a significant *p*-value ≤ 0.05 were annotated by gene ontology (GO) according to [54] and [55], and KEGG pathway annotation according to Kanehisa et al. [56].

### 2.6. qRT-PCR Analysis

#### 2.6.1. Screening of Candidate Reference Genes and Primer Design

A total of 4 candidate reference genes were evaluated from http://www.icugi.org/ (accessed on 9 April 2017). These genes were chosen based on previous work by Guo et al. [57] and included: PAL, LOX1, CTR1, and beta-actin. All gene primers for qRT-PCR, except beta-actin, which was referenced by Kong et al. [49], were designed by Beacon Designer 7. All primers were synthesized by Quintarabio, Wuhan, China. The specificity of the PCR amplification product for each primer pair was further studied by melting curve analysis and amplification efficiency (E). All reactions were performed in a 20-μL volume containing 10 μL 2 × UltraSYBR Mixture (Cwbio Biotech Co., Ltd., Beijing, China), 1μL 10Μm forward and reverse primers, and 500 ng of cDNA. Melting curve analysis and amplification efficiency were conducted using Bio-Rad CFX Manager™ Software on Bio-Rad iQ5 real-time PCR instruments (Bio-Rad Laboratories, Hercules, CA, USA). The PCR program was as follows: 95 °C for 1 min, 40 cycles of 95 °C for 10 s, 60 °C for 10 s, and 72 °C for 40 s. Finally, dissociation curves were generated by increasing the temperature from 65 to 95 °C.

#### 2.6.2. Total RNA Extraction and cDNA Synthesis

Total RNA was extracted from root samples using the Biozol method (Biomiga, Shanghai, China). Complementary DNA (cDNA) synthesis was then performed using a PrimeScript™ RT Reagent Kit with gDNA Eraser (Takara Biotechnology Co., Ltd., Dalian, China) according to the manufacturer’s instructions. The primer pairs for quantitative RT-PCR were designed using Beacon Designer (Premier Biosoft, Palo Alto, CA, USA). The PCR reaction consisted of 10 μL of 2× Ultra SYBR Mixture (Cwbio, Beijing, China), 40 nM primers, and 2 μL of 1:10 diluted template cDNA in a total volume of 20 μL. No template controls were set for each primer pair. Quantitative RT-PCR was performed employing the Bio-Rad CF X96 System and Opticon Monitor software (Bio-Rad, USA). CT values were determined by averaging three technical replicates and three biological replicates. For each gene of interest, standard curves were generated. We converted Ct values to relative amounts of cDNA and calculated gene expression using the 2^−ΔΔCT^ method [58].

#### 2.6.3. Quantitative Determination and Statistical Analysis

The expression levels of the tested reference genes were determined by CT values. The value of E for each reference gene was calculated according to the following equation: E (%) = (10^−1/slope^ − 1) × 100, where the slope is the standard curve slope determined using Excel 2016. The calculations and comparisons of treatment means for each experiment were conducted using analysis of variance (ANOVA) and SPSS 20.0 (IBM, Armonk, NY, USA); means were analyzed with Tukey’s honestly significant difference (HSD) test at *p* = 0.05.

### 2.7. Data Analysis

All data were analyzed using SPSS 20.0 (IBM, Armonk, NY, USA) and Excel 2016 (Microsoft, Redmond, WA, USA). The results of qRT-PCR were calculated by REST using the 2^−ΔΔCT^ method, and all figures were generated using GraphPad Prism 6 (GraphPad Software, San Diego, CA, USA).

### 2.8. MS/MS Data Submission

The raw data used in the present study for proteomics were submitted to the Integrated Proteome Resources database under accession number, IPX0001180000.

## 3. Results

### 3.1. The Control Efficiency of F1-35 against Watermelon Fusarium Wilt

To test the efficiency of F1-35 in controlling watermelon Fusarium wilt, F1-35, F1-35, and FON were simultaneously inoculated, and the incidence of watermelon seedling infections was recorded. The resulting control efficiency of F1-35 on watermelon Fusarium wilt was 61.7% (Table 1).

### 3.2. Pairing Assay

FON grew synchronously in the presence of F1-35, with no reduction compared to FON only. The radius of the FON colony was 28.5 mm and 27.7 mm, respectively, with no significant difference by the *t*-test. Therefore, F1-35 did not change the growth rates of FON. There was a boundary between F1-35 and FON. Subsequently, the mycelia of FON, which was located beside F1-35 colony, were observed with a microscope. There were no changes in either the mycelia or the spores. As such, no direct interactions between FON and F1-35 were observed under the microscope.

### 3.3. Influence of the Root Proteome Expressed after F1-35 Inoculation

To investigate the mechanism of Fusarium wilt reduction under F1-35, watermelon root proteomes during treatment with F1-35 (samples collected at 12, 24, and 48 hpi) were compared to CK. A total of 31,440 peptides were identified, including 27,618 unique peptides, and 6243 proteins were blasted to Benincaseae. To establish authenticity and accuracy, the MASCOT ion score was set as the index, which was carried with MASCOT 2.2 to form a blast in the database. The result of MASCOT showed that 70% of total peptides ion scores were higher than 20, with the middle ion score being 30. Comparing the FDR < 0.01 and ion score distribution, the MS result appears to be reliable and reproducible.

Proteomic results were summarized in the Venn diagram in Figure 1. This clearly demonstrated the differently expressed proteins in watermelon response to F1-35 at 12, 24, and 48 hpi. For the 12-hpi root proteins compared to CK, there were 184 differentially expressed proteins, 110 with up-regulated expression and 83 with down-regulated expression. For the 24-hpi root proteins, the differentially expressed proteins increased to 412, with 221 up-regulated and 191 down-regulated. The 48-hpi root differentially expressed proteins were similar to the 24-hpi root proteins, where 413 proteins were differentially expressed, 186 up-regulated and 227 down-regulated.

To investigate the interaction between F1-35 and watermelons, the identified proteins were analyzed. Compared to CK at 12 hpi, there were three up-regulated proteins and three down-regulated proteins. Having blasted the function of those proteins, the three up-regulated proteins related to soil stress or drought stress, and the three down-regulated proteins were Gip1-like proteins, which respond to some stress. Auxin-repressed protein (Arp), induced by JA, could increase resistance after being silenced. Compared to CK at 24 hpi, the highly expressed protein after treatment for 24 h with F1-35 had not only GIP, Arp, and the negative-regulated resistance protein, but also some JA-associated proteins, PR5 and glyceraldehyde-3-phosphate dehydrogenase. The down-regulated proteins were similar to the 12-hpi samples. After treatment for 48 h with F1-35, the major differentially expressed proteins were similar to 12 h and 24 h. The down-regulated proteins were WRKY 17, a negative protein to JA, and a negative-regulated resistance protein (Table 2).

### 3.4. Identification and Functional Classification of Differentially Expressed Proteins

GO analysis revealed that most differentially expressed proteins were involved in metabolic processes, and single organism and cellular processes, among others. Many differentially expressed proteins were associated with molecular functions, such as binding, catalytic activity, and the binding of small molecules. Moreover, many differentially expressed proteins were associated with the cellular component, such as the organelle, membrane, and cell part, as seen in (Figure 2).

The differentially expressed proteins were also grouped into main functional classes based on KEGG [59]. The KEGG pathway analysis showed that most differentially expressed proteins occurred in plant–pathogen interaction, phenylpropanoid biosynthesis, oxidative phosphorylation, carbon metabolism, alpha-linolenic acid metabolism, cysteine and methionine metabolism, and the MAPK signaling pathway to the plant (Figure 3). In the root proteins at 12 and 24 hpi, the key enzymes were identified in alpha-linolenic acid metabolism and cysteine/methionine metabolism pathways. The plant hormones, JA and ET, are the product of those pathways. In the 24- and 36-hpi root proteins, JAR1 and ChiB were identified.

Based on the above results, we consider that the mechanism of F1-35 protection watermelon against Fusarium wilt involves triggering the JA and ET pathways (Figure 4).

### 3.5. qRT-PCR Analysis

To validate the above hypothesis, qRT-PCR was used. The primers were designed with Beacon Designer 7 according to the cDNA sequence, which can be downloaded from http://www.icugi.org/ (accessed on 9 April 2017), with the exception of beta-actin (Table 3). To determine a suitable temperature for qRT-PCR, a gradient PCR was carried out, with results shown for 60 °C; each primer had a well-defined peak. The results of E show that all primers had amplification efficiency values of 96.28 to 224.35%, and all R^2^ values were greater than 0.96, all of which indicated that those primers could be used in qRT-PCR and 2^−ΔΔCT^ [58].

Together, these results suggest that all primers were suitable for qRT-PCR. Therefore, samples collected after treatment for 0, 12, 24, 48, 72, and 96 h were used to analyze gene expression, including the PAL, LOX1, and CTR1 genes. In F1-35 treated roots, PAL gene transcriptional levels were also induced after 48 hpi, whereas in FON-treated roots, the PAL gene was highly expressed before 48 hpi. In F1-35-treated stems, the PAL gene was induced at 48 hpi, and peaked at 72 hpi, which was higher than in the CK or FON treatments. In F1-35-treated leaves, the PAL gene was induced at 24 hpi. The peak of the PAL gene was 5.34 at 24 hpi, and then it slowly returned to normal (1.34). With FON treatment, PAL gene expression was increased after an initial decrease. The LOX1 gene was induced after treatment with F1-35 at 12 hpi, and peaked at 24 hpi in the roots. In stems and leaves, LOX1 was also induced. CTR1, a negative key gene connected with the ET pathway, was down-regulated after treatment with F1-35 and even FON (Figure 5).

## 4. Discussion

*Fusarium oxysporum* f. sp. *niveum* (FON), the pathogen causing watermelon Fusarium wilt, is wildly distributed around the world, and *F. oxysporum* is a hemi-biotrophic pathogen that can infect many plants, including tomatoes, bananas, and *Arabidopsis* [60]. FON poses a serious threat to the yield and quality of watermelons. A biocontrol agent is an effective natural means of controlling plant disease. All BCAs have been isolated from the rhizosphere and plants, but there are some obstacles in the application of BCAs. One of the important factors is the loss of anti-fungal activity, a result of a longtime subculture on artificial media [61,62,63,64]. Several methods can resolve this loss, including protoplast fusion and molecular genetics. As a Fo47-actinomycetes-153 fusion strain, previous research found that F1-35 produce more gibberellin and have a higher control efficiency of FON in watermelon plants, up to 59%, which was higher than Fo47 [48]. Our control efficiency experiment demonstrated a 61.7% control effect of FON in watermelon seedlings. Indeed, Fo47, improved with actinomycetes 153 by protoplast fusion, had stable biocontrol activity.

Further, the mechanism of F1-35-controlled Fusarium wilt in watermelon seedlings was explored. Previous studies reported that Fo47 controls plant disease by nutrition and ecological niche competition, inducing the expression of some defense genes. Actinomycetes 153 has activity against some fungi and bacteria [48]. As such, after protoplast fusion, did the Fo47-actinomycetes-153 fusion strain have those abilities? In this study, pair assaying was used to explore the antifungal ability of Fo47 after improvement with the protoplast of actinomycetes 153. The pair assay results showed that F1-35 could not cause the inhibition of FON growth because both the mycelia growth rates and spore morphology were normal.

In plants, *F. oxysporum* is controlled by manipulating defense pathways: some research has indicated that *Arabidopsis thaliana* responds against *F. oxysporum* via the JA/ET pathway [65]. Some studies have reported that SA and JA biosynthesis genes can be induced to expression at the early stage of FON infection of watermelons, and a lower concentration of SA may enhance watermelon resistance to FON. In addition, in resistant watermelon varieties, the crosstalk net between SA, JA, and ABA may help the JAR, NPRs, and PYLs family genes trigger the plant immune system against FON infection [22,66]. Other studies reported that *F. oxysporum* resistance is independent of SA-dependent defense gene expression in impaired JA-signaling mutants, such as myc2, coi1, and pft1/med25 [67,68,69]. In our study, the improved control agent, F1-35, increased some key enzymes of JA and ET pathways, which is similar to other BCAs, such as *Bacillus pumilus* SE34 [70,71]. Moreover, our results also indicate that F1-35 not only stimulates the JA and ET pathways, but also triggers plant PAMP defense and many broad-spectrum resistance proteins. Compared to CK treatment at 12 hpi, the up-regulated proteins were Q6I673, A3F570, and P0DI61. Former research has shown that these are a response to a poor soil environment and drought stress [72,73]. The down-regulated proteins were A0A0A0LG20, D1MWZ6, and A0A0A0L4M8. A0A0A0LG20 is a gibberellin-binding protein. D1MWZ6 is an auxin-repressing protein and A0A0A0L4M8 is a negative-regulated resistance protein [74]. Compared to CK treatment at 24 hpi, the up-regulated proteins were V5LF72, A0A097BU00, Q6I673, A0A0A0LWG8, A0A1D8RFV5, A0A0A0LW70, A0A0A0K1Y0, H6TB43, A0A0A0KGB3, A5X4I4, and A0A0A0KBB7. The main function of these proteins is drought resistance, including the PR5 protein or connecting with the JA pathway. The down-regulated proteins were D1MWZ6 and A0A0A0L4M8, which is similar to the result of 12-hpi vs. CK treatment. Compared to CK treatment at 48 hpi, the up-regulated proteins were Q6I673, A0A0A0LVN5, and A0A0A0LD49. These are connected to the JA pathway and some resistance proteins. The down-regulated proteins were E7CEW6, A0A0A0L4M8, A0A0A0LT35, D1MWZ6, and A0A0A0LW64. These proteins are negative-regulated resistance proteins or have a negative correlation with JA.

Furthermore, plant hormones were systemically analyzed. JA and ET, as plant hormones, play important roles in plant life, including fruit-ripening, and responding to pathogen and environment stress [75]. The signal transduction pathways occur in the induced responses of a plant to the pathogen, wounding, and herbivory. Jasmonic acid is involved in inducible defenses against pathogens and insects, and ET is important in induced plant defense [76,77].

In this study, lipoxygenase (LOX) and allene oxide synthase (AOS), key enzymes in the JA-producing pathway [78], were identified. Indeed, jasmonic acid-amido synthetase (JAR1), a JA-induced protein [79], could also be identified. In addition, chitinase B (ChiB), a protein related to downstream ethylene signal transduction [80], was elevated. Therefore, the JA and ET pathways were induced after the inoculation of F1-35 in the watermelons. To validate our results, qRT-PCR was used to analyze *PAL*, *LOX1*, and *CTR1* gene expression levels. The gene expression results show that *PAL* and *LOX1*, which have a positive correlation to JA, have up-expression, and the negative-associated gene, *CTR1*, was down-expressed. In conclusion, the improved biocontrol agent, F1-35, can improve the resistance of watermelons to FON by triggering the JA and ET pathways.

## Figures and Tables

**Figure 1 microorganisms-10-01710-f001:**
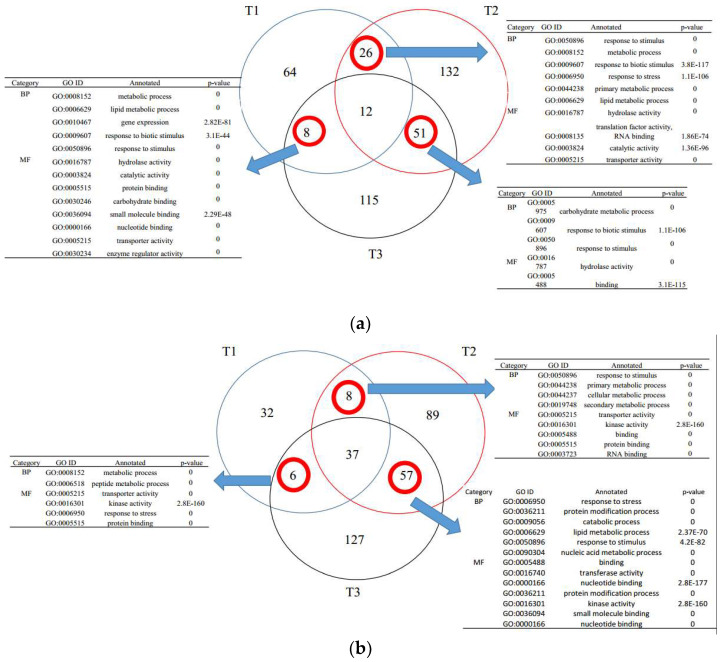
Distribution of differentially expressed proteins. (**a**): The up-regulated expression in watermelon response to F1-35 at 12, 24, and 48 hpi. (**b**): The down-regulated expression in watermelon response to F1-35 at 12, 24, and 48 hpi. **T1:** CK treatment at 12 hpi; **T2**: CK treatment at 24 hpi; **T3**: CK treatment at 48 hpi.

**Figure 2 microorganisms-10-01710-f002:**
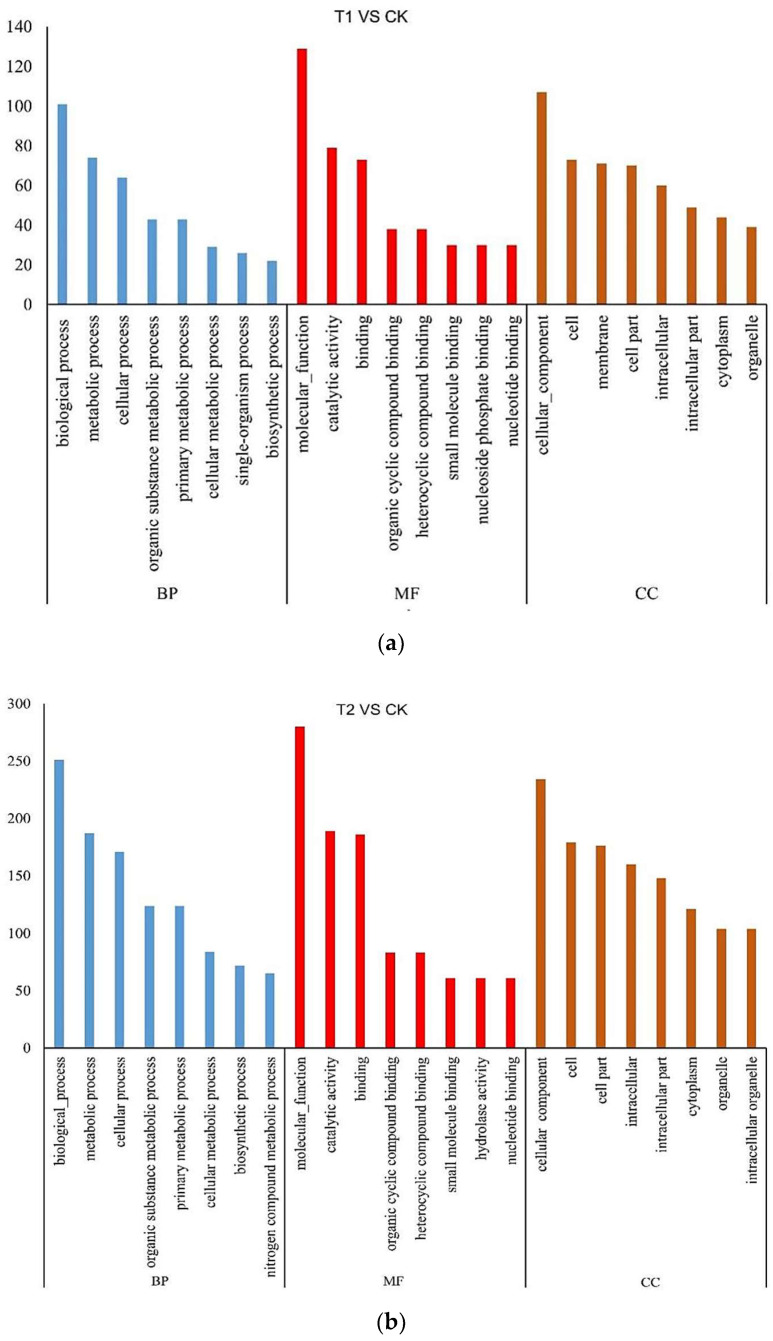
Gene ontology annotation. (**a**): CK treatment at 12 hpi; (**b**): CK treatment at 24 hpi; (**c**): CK treatment at 48 hpi. **BP:** biological process; **MF:** molecular function; **CC:** cellular component.

**Figure 3 microorganisms-10-01710-f003:**
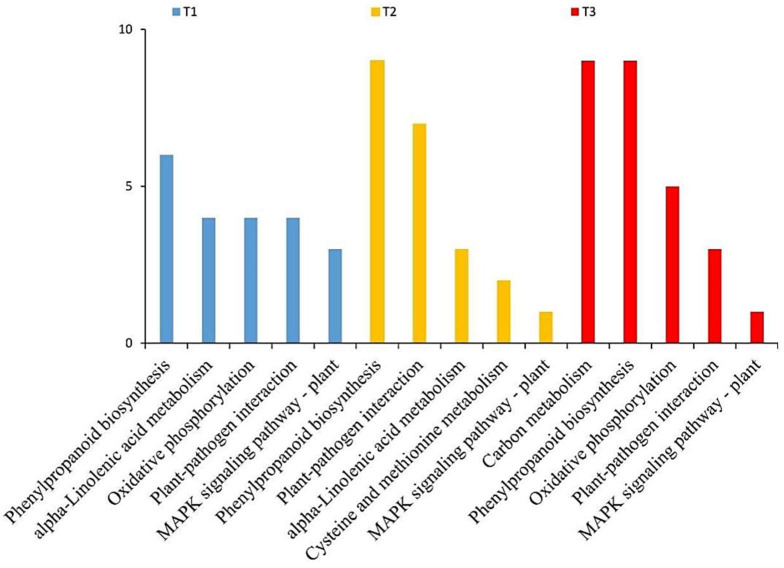
KEGG annotation. T1 is CK treatment at 12 hpi; T2 is CK treatment at 24 hpi; and T3 is CK treatment at 48 hpi.

**Figure 4 microorganisms-10-01710-f004:**
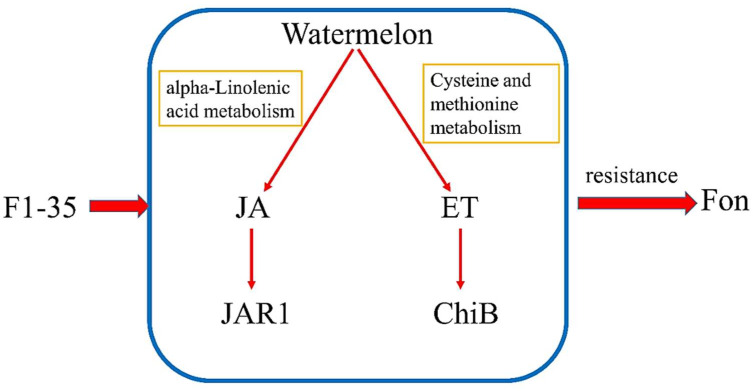
A summary of the mechanism of F1-35 protection of watermelons against FON.

**Figure 5 microorganisms-10-01710-f005:**
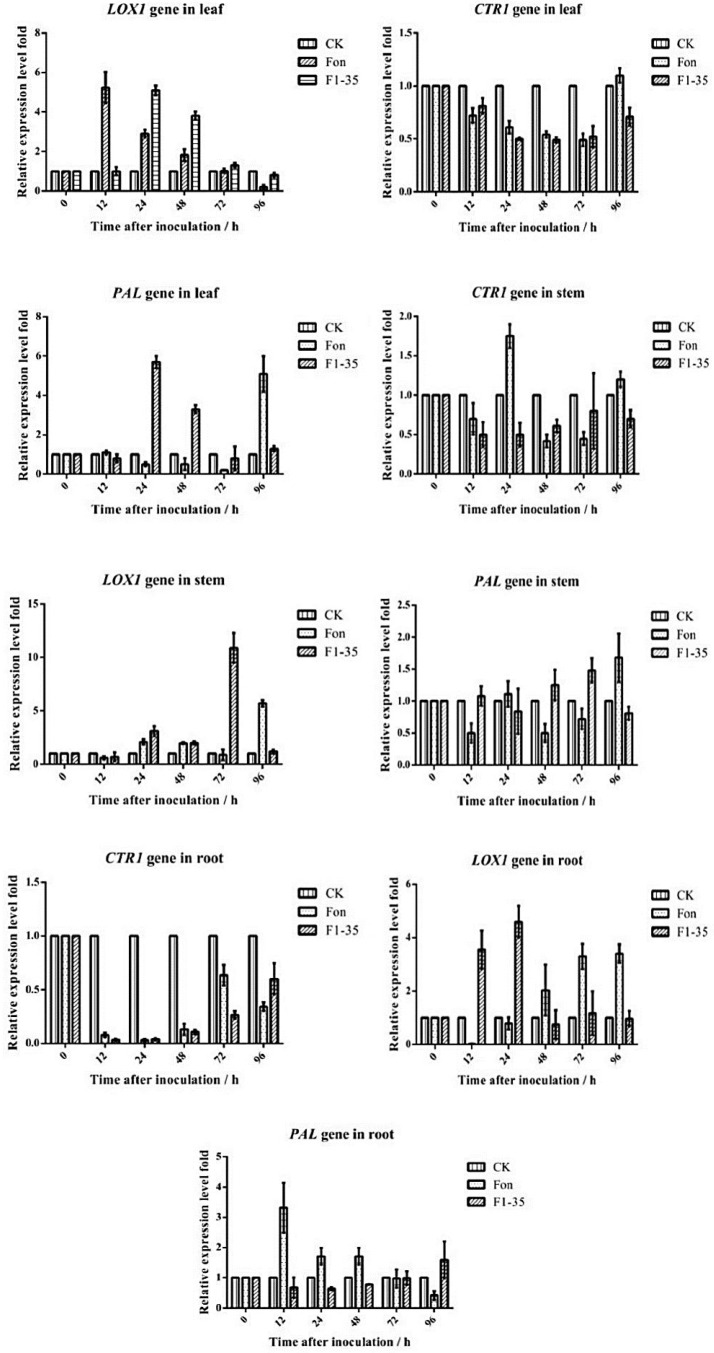
Target gene expression level in the roots, stems, or leaves of plants treated with F1-35 or FON.

**Table 1 microorganisms-10-01710-t001:** Control efficiency of F1-35 on watermelon Fusarium wilt.

Treatment	Disease Index (%)	Control Efficiency (%)
FON	58.46 ± 4.48	-
F1-35 and FON	22.39 ± 5.01	61.7 ± 3.79

**Table 2 microorganisms-10-01710-t002:** Function analysis of differentially expressed proteins.

12 hpi /CK				24 hpi /CK				48 hpi /CK			
Up	Function	Down	Function	Up	Function	Down	Function	Up	Function	Down	Function
Q6I673	Drought stress	A0A0A0LG20	Drought stress	V5LF72	Broad-spectrum resistance	D1MWZ6	Arp	Q6I673	Broad-spectrum resistance	E7CEW6	WRKY17
A3F570	Drought stress	D1MWZ6	Drought stress	A0A097BU00	Alcohol dehydrogenase	A0A0A0L4M8	Susceptibility	A0A0A0LVN5	Alcohol dehydrogenase	A0A0A0L4M8	Susceptibility
P0DI61	Soil stress	A0A0A0L4M8	Soil stress	Q6I673	Drought stress			A0A0A0LD49	Drought stress	A0A0A0LT35	Susceptibility
				A0A0A0LWG8	Related to JA					D1MWZ6	Arp
				A0A1D8RFV5	PR5					A0A0A0LW64	Negative correlation to heat shock
				A0A0A0LW70	Physiological stress						
				A0A0A0K1Y0	Ca						
				H6TB43	Osmotic stress						
				A0A0A0KGB3	Related to JA						
				A5X4I4	glycerol 3-phosphate degradation						
				A0A0A0KBB7	Precursors of JA						

**Table 3 microorganisms-10-01710-t003:** The primers for qRT-PCR.

Gene Name	Gene ID	Primers		Production Size (bp)	Amplification Efficiency %	R^2^
β-actin	MU51303	F	CCTGGTATCGCTGACCGTAT	133	96.7	
		R	TACTGAGCGATGCAAGGATG			
*PAL*	Cla018297	F	TGCTATGGCTTCCTATT	141	114.35	0.9912
		R	ATGTCAATGGCTTCTTC			
*LOX1*	Cla019905	F	AATGCTTGCTGGAGTGA	121	99.25	0.9946
		R	TGCTATGTGTTCTTCTGTTATG			
*CTR1*	Cla017731	F	GAAGTTGCTGTGAAGAT	100	96.28	0.9656
		R	TAGGATGTCGTAAGGATT			

Note: beta-actin data taken from Kong et al. [49].

## Data Availability

The raw data used in the present study for proteomics were submitted to the Integrated Proteome Resources database under accession number IPX0001180000.

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
