# Peer review of "The Improved Biocontrol Agent, F1-35, Protects Watermelon against Fusarium Wilt by Triggering Jasmonic Acid and Ethylene Pathways"

_microorganisms, 2022, doi:10.3390/microorganisms10091710_

Round 1

Reviewer 1 Report

The Figures 2 and 3 and Table 2 were difficult to relate to the presented results. BP, MP, and CC were not defined in Figures. These figures were providing Gene ontology annotation for the check treatment only, but in the results, the authors listed the differences associated with the biocontrol agent compared to the check.  Is there no way to put that information into a figure or table, rather than concentrating on only the check in your figure 2 and 3?  The discussion presented better details on the results concerning Table 2, than did the results section, though in some cases, the differentially expressed proteins in Table 2 were not the same presented as in the discussion.   Table 2 is poorly presented and should be redone so that it is easier to follow.  In the literature section, number 15 should be Meloidogyne incognita, with both words in italics. Number 72, the l in lycopersici should be lower case.

Reviewer 2 Report

In the present study, the authors have studied the efficacy of biocontrol agent F1-35 against watermelon Fusarium wilt. Results of this study showed biocontrol agent F1-35 improves the resistance of the watermelon plant by triggering JA and ET pathways. The manuscript is well written except for some minor grammatical errors. The findings of this study corroborated earlier reports. However, the discussion section of the manuscript is written poorly. It should be discussed concerning the earlier watermelon and FON interaction findings.  

Following some comments, authors can consider improving the manuscript draft.

1.       Introduction: page 1, line 33 and 35- Correct “Watermelon Fusarium wilt” as “Fusarium wilt of watermelon”

2.       Introduction: page 1, line 35-37- Rewrite the sentence in the context of pathogen which causes wilt in watermelon.

3.       Introduction: page 2, line 49- change “Fusarium oxysporum” to “F. Oxysporum”

4.       Introduction: page 2, line 49- Add some recent literature findings related to FON and watermelon interaction through JA. Please refer article by Kasote et al., 2020, BMC plant biology.

5.       Material and methods: page 3, line-99- Change “varietsy” to “variety”.

6.       Merge section 2.5.3. and 2.5.4.

7.       Material and methods: page 5, line-206- Correct “65°C to 95°C” to “65-95°C

8.       Discussion: page 13, line 346- delete repeated “Fusarium oxysporum”

9.       Material and methods: page 5, line-250- check “plant et al.,?? 
